# Formation and Investigation of Physicochemical, Biological and Bacteriostatic Properties of Nanocomposite Foils Containing Silver Nanoparticles and Graphene Oxide in Hyaluronic Acid Matrix

**DOI:** 10.3390/ma14123377

**Published:** 2021-06-18

**Authors:** Karen Khachatryan, Lusine Khachatryan, Marcel Krzan, Magdalena Krystyjan, Lidia Krzemińska-Fiedorowicz, Anna Lenart-Boroń, Aneta Koronowicz, Mariola Drozdowska, Gohar Khachatryan

**Affiliations:** 1Faculty of Food Technology, University of Agriculture in Krakow, Balicka Street 122, 30-149 Krakow, Poland; magdalena.krystyjan@urk.edu.pl (M.K.); lidia.krzeminska-fiedorowicz@urk.edu.pl (L.K.-F.); aneta.koronowicz@urk.edu.pl (A.K.); mariola.drozdowska@urk.edu.pl (M.D.); gohar.khachatryan@urk.edu.pl (G.K.); 2Clinical Department of Orthopedics and Traumatology, University Hospital in Krakow, Macieja Jakubowskiego 2, 30-688 Kraków, Poland; lusine93@gmail.com; 3Jerzy Haber Institute of Catalysis and Surface Chemistry, Polish Academy of Sciences, Niezapominajek 8, 30-239 Krakow, Poland; 4Faculty of Agriculture and Economics, University of Agriculture in Krakow, Mickiewicz Ave. 21, 31-120 Krakow, Poland; anna.lenart-boron@urk.edu.pl

**Keywords:** silver nanoparticles, graphene, hyaluronic acid, bacteriostatic properties, cytotoxic effect

## Abstract

Natural polysaccharides, including hyaluronic acid, find a wide range of applications in biomedical sciences. There is a growing interest in nanocomposites containing hyaluronic acid and nanoparticles such as nanometals or graphene. In this study, we prepared foils of pure sodium hyaluronate and sodium hyaluronate containing nanosilver, graphene oxide, nanosilver/graphene oxide and characterized their properties. UV-vis spectroscopy and scanning electron microscopy (SEM) confirmed the formation of 10–20 nm silver nanoparticles. The structural changes were investigated using Fourier transforms infrared (FTIR) spectra and size exclusion chromatography. The obtained results suggest changes in molecular weights in the samples containing nanoparticles, which was highest in a sample containing nanosilver/graphene oxide. We also assessed the mechanical properties of the foils (thickness, tensile strength and elongation at break) and their wettability. The foils containing nanosilver and nanosilver/graphene oxide presented bacteriostatic activity against *E. coli*, *Staphylococcus* spp. and *Bacillus* spp., which was not observed in the control and sample containing graphene oxide. The composites containing graphene oxide and nanosilver/graphene oxide exhibited a cytotoxic effect on human melanoma WM266-4 cell lines (ATCC, Manassas, VA, USA).

## 1. Introduction

Polysaccharides are commonly available and biodegradable raw materials attractive to various industries (such as chemical or food technology, medicine, pharmaceutics). The interest in natural polysaccharides is increasing due to their expanding potential applications [1]. The key parameters for their practical use are low, medium, and high molecular weights, variable polydispersity, forming linear and branched macrostructures, monofunctionality, high degree of chirality, either low or high aqueous solubility, low (if any) toxicity, and immunogenicity. The bioactivity of polysaccharides and their antioxidative, immunomodulating, anti-inflammatory, antiviral, antimutagenic, cancerostatic and anticlotting properties motivate their application in nanotechnology, medicine, and food technology [2,3,4,5,6]. In recent years, various polysaccharides have been used in the synthesis of inorganic nanoparticles, acting as reducers and stabilizers and providing the formation of nanoparticles of uniform size [7,8,9,10]. The following polysaccharides are suitable for developing nanoparticles: starch, cellulose, alginates, pectins, xanthan gum, cyclodextrins, chitosan, heparin, and hyaluronic acid [11,12,13].

Hyaluronic acid is a polymer of D-glucuronic acid and N-acetyl-D-glucosamine that is widely spread in nature. Although it resides mainly in intercellular matrices of vertebrates, it can also be found in the organisms of certain bacteria (*Aerobacter aerogenes* and *Streptococcus pyogenes*] [14]. It also resides in the human organism as a component of so-called Wharton’s jelly, skin, synovial fluid, and the eye’s vitreous. Hyaluronic acid plays an essential role in embryo development, tissue growth, angiogenesis, wound healing, and controls the biomechanical properties of tissues. For these properties, hyaluronic acid has found several applications in tissue engineering, cosmetology, and various fields of medicine. Because of its anionic character, hyaluronic acid has a remarkable water binding capacity and is suitable for hydrogel production. In vitro studies showed that this polysaccharide positively influences cell migration and building extracellular substances. Depending on the molecular mass and additional functional groups it can either stimulate or inhibit various paths and processes within living organisms, including those associated with tumors [15].

Graphenes and their derivatives are single-layered carbon materials. The large surface area on both sides of the sheet is provided by their unique 2D structure and enables physical adsorption of nucleobases and aromatic compounds, mainly through π–π stacking. Various functional groups with different polarities and ionic characters (hydroxyl, epoxy and carboxyl groups) present in graphene oxide (GO) sheets give an opportunity to create new materials by interactions with biomolecules [16]. Recently, they have attracted considerable interest from biomedical researchers.

Although GO is monodisperse in pure water, its aggregation occurs in highly concentrated solutions of salts or proteins, for example, culture medium and serum. Different functional groups have been added to GO to make it more stable and biocompatible in the aqueous phase. Hydrophilic groups, such as poly(ethylene glycol) sulfonic acid groups and polysaccharides, make GO more promising in many fields, for example, in drug delivery [17,18,19,20,21,22].

Jung et al. successfully prepared a nanographene oxide–hyaluronic acid (NGO-HA) conjugate. Because of its structure and properties, it could be used to target specific drug delivery via HA-receptor-mediated endocytosis. The pH-dependent drug release and target specific anti-cancer effect was confirmed by in vitro tests [23]. Another study, performed by Song et al. [24] presented the synthesis and properties of a novel nanohybrid of hyaluronic acid (HA)-decorated graphene oxide (GO), fabricated as a targeted and pH-responsive drug delivery system. The obtained nanohybrid was successfully tested for controlling the release of the anti-cancer drug doxorubicin (DOX) and it was confirmed that the HA−GO−DOX nanohybrids have potential clinical applications for anti-cancer drug delivery. Dantas et al. prepared a nanohybrid composed of GO functionalized with HY; it was tested in vitro and in vivo and proved to accelerate the tissue regeneration process in bone defects created in the tibia of rats [25].

Graphene, as other nanoparticles, exhibits a tendency to aggregation via the Van der Waals interaction between its layers and such behavior limits its applications. Metal nanoparticles introduced onto the graphene surface prevent aggregation and provide good photoconductivity and catalytic properties of the graphene/metal nanocomposites [26]. Good conductivity, strong ultraviolet-visible absorption ability, and catalytic activity of silver nanoparticles are why they are popular in many fields, such as for use in electronic devices, as biomarkers, and as antibacterial agents [27]. Recently, the interest in graphene/metal nanoparticle composites has been growing, especially in catalysis, energy storage, chemical sensors, and hydrogen storage [28].

Polymeric nanocomposites with nanosilver particles sized from 1 to 100 nm evoke considerable interest in biomedical sciences. Depending on the size of those particles, that is, their surface to volume ratio, they exhibit different biological, chemical, and physical properties [29]—among them, the most essential for biological sciences are bactericidal and fungicidal properties. The morphology and stability of silver nanoparticles depend on the mode of their preparation [30], applied reducer, and stabilizing reagent.

Due to the emergence and increase in the number of antimicrobial resistant, as well as multidrug resistant, microorganisms [31], including typically environmental species of bacteria [32], new efficient and cost-effective antimicrobial agents that would overcome this resistance are sought [31]. Nanoparticles are now considered a reasonable alternative to antimicrobial agents and seem to have the potential to solve the problem of multidrug resistance among bacteria [33]. A number of studies have been published to date, demonstrating the antimicrobial potential of silver nanoparticles obtained using various methods [34,35,36], thus making them promising agents not only for fighting infections but in many other biomedical areas [33]. The studies performed by Khachatryan et al. [37,38] presented the green synthesis of silver nanoparticles using hyaluronan as a stabilizing template. The obtained composites exhibited bacteriostatic activity, water solubility, and promising physicochemical and functional properties.

In this study, we prepared four types of foils containing pure sodium hyaluronate (Hyal/C), nanosilver in Hyal matrix (Hyal/Ag), graphene oxide in Hyal matrix (Hyal/GO) and nanosilver with graphene oxide in Hyal matrix (Hyal/Ag/GO) using environmentally-friendly methods. We also characterized their morphologies, and physicochemical and biological properties.

## 2. Materials and Methods

### 2.1. Materials

Sodium hyaluronate (Sigma-Aldrich, Poznan, Poland, PubChem CID: 3084049), AgNO_3_ (Aldrich, Poznan, Poland, 99.99%, PubChem CID: 24470), NH_3_ (Sigma-Aldrich, Poznan, Poland, PubChem CID: 18944693) and D-(+)-xylose (Sigma-Aldrich, Poznan, Poland, BioXtra, PubChem CID: 135191) and deionized water were applied for the preparation of composites with nano Ag.

### 2.2. Preparation of Nanoparticles Composites

#### 2.2.1. Preparation of Hyal/C Foils

Aq. sodium hyaluronate (0.8 g in 80 mL water) was maintained at 30 °C under continuous stirring. The resulting homogeneous and transparent gel was divided into two equal parts—one half (for the assessment of mechanical, structural and antibacterial properties) was applied to a clean polypropylene surface and dried at 50 °C to constant weight and the dry foils were then collected and stored in closed vessels. The remaining gel was used for the cytotoxicity tests.

#### 2.2.2. Preparation of Hyal/Ag Composite

The nanocomposite was prepared according to the described procedure [37]. Aq. sodium hyaluronate (0.8 g in 80 mL water) was maintained at 30 °C under continuous stirring. Aq. AgNO_3_ solution (1.5 g in 100 mL water) (0.6 mL) was then introduced followed by addition of 4% *w*/*w* aq. ammonia (2 mL), 4% *w*/*w* aq. xylose (8 mL). The reaction mixture was then agitated for 30 min with magnetic stirrer at 60 °C in a thermostatic water bath. The resulting suspension of nanoAg in the Hyal matrix was lowered to room temperature and then centrifuged (5000 rpm). The gel was divided into two equal parts—one half (for the assessment of mechanical, structural, and antibacterial properties) was applied to a clean polypropylene surface and dried at 50 °C to constant weight and the dry foils were collected and stored in closed vessels. The remaining gel was used for the cytotoxicity tests.

#### 2.2.3. Preparation of Hyal/GO Composite

Graphene oxide (GO) was synthesized using the modified Hummers method [39]. Aqueous suspensions containing 0.1% of GO in distilled water were prepared and treated using ultrasounds.

Sodium hyaluronate powder (0.8 g) was added to a deionized then distilled water (80 mL) and maintained at 30 °C under constant stirring. After stirring the reaction mixture, 4 mL of the suspension of GO (0.1%) was added and then mixed for 2 h, followed by cooling to room temperature and then it was centrifuged (5000 rpm). The gel was divided into two equal parts—one half (for the assessment of mechanical, structural and antibacterial properties) was applied to a clean polypropylene surface and dried at 50 °C to constant weight and the dry foils were collected and stored in closed vessels. The remaining gel was used for the cytotoxicity tests.

#### 2.2.4. Preparation of Hyal/Ag/GO Composite

Sodium hyaluronate powder (0.8 g) was added to a deionized then distilled water (80 mL) and maintained at 30 °C under constant stirring. Aq. AgNO_3_ solution (1.5 g in 100 mL water) (0.6 mL) was introduced followed by the addition of 4% *w*/*w* aq. ammonia (2 mL), 4% *w*/*w* aq. xylose (8 mL). That reaction mixture was then agitated for 30 min with a magnetic stirrer at 60 °C in a thermostatic water bath. After stirring the reaction mixture, 4 mL of the suspension of GO (0.1%) was added and then mixed for 2 h, followed by cooling to room temperature. The gel was divided into two equal parts—one half (for the assessment of mechanical, structural and antibacterial properties) was applied to a clean polypropylene surface and dried at 50 °C to constant weight and the dry foils were collected and stored in closed vessels. The remaining gel was used for the cytotoxicity tests.

The resulting flexible and transparent films are shown in Figure 1.

### 2.3. Characterization of Nanoparticle Composites

#### 2.3.1. Determination of Molecular Weight MW and Radii of Gyration Rg of Starch Polysaccharide Molecules

A high-performance size exclusion chromatography (HPSEC) system with a multiangle laser light scattering (MALLS) detector and a differential refractive index detector (RI) were used to determine the molecular weight and radii of gyration of the starch polysaccharide molecules. The high-performance size exclusion chromatography (HPSEC) system consisted of a pump (Ultimate 3000, Dionex, Palo Alto, CA, USA), an injection valve (model 7021, Rheodyne, Palo Alto, CA, USA), a guard column (TSK PWH, Tosoh Corporation, Tokyo, Japan) and two connected size exclusion columns: TSKgel GMPWXL (300 mm × 7.8 mm, Tosoh Corporation, Tokyo, Japan) and TSKgel 2500 PWXL (300 mm × 7.8 mm, Tosoh Corporation, Tokyo, Japan). A multiangle laser light scattering (MALLS) detector (Dawn-DSP-F, Wyatt Technology, Santa Barbara, CA, USA) and a differential refractive index detector (model SE71, Shodex, Tokyo, Japan) were connected to the columns.

The flow rate of the mobile phase and the sample injection volume were 0.4 mL · min^−1^ and 500 μL, respectively. Calculation of the weight-average molecular weight (Mw) and radius of gyration (Rg) were performed using Astra 4.70 software (Wyatt Technology, Santa Barbara, CA, USA).

#### 2.3.2. UV-Vis Absorption Spectrophotometry

The UV-vis absorption spectra of the composites was recorded using a Shimadzu 2101 scanning spectrophotometer in the range of 200–700 nm using 10 mL cells (Hellma Materials GmbH, Jena, Germany). Concentration of solution was 0.001 g/mL.

#### 2.3.3. Scanning and Transmission Electron Microscopy (SEM and TEM)

The morphology of as-prepared nanocomposites was studied using a high resolution JEOL JSM—7500 F (Akishima, Tokyo, Japan) field emission scanning electron microscope equipped with transmission electron detector (TED) and a retractable backscattered-electron detector (RBEI).

#### 2.3.4. FTIR-ATR Spectrophotometry

The FTIR-ATR spectra of the composites were recorded in the range of 4000–700 cm^−1^ using a MATTSON 3000 FT-IR (Madison, WI, USA) spectrophotometer. The instrument was equipped with a 30SPEC 30° reflectance adapter fitted with the MIRacle ATR accessory from PIKE Technologies Inc., Madison, WI, USA.

#### 2.3.5. Thickness and Mechanical Properties of Composites

The measurements were undertaken by using a micrometer, catalog no.: 805.1301 (Sylvac SA, Crissier, Switzerland), with a 0.001-mm resolution. The thickness was expressed as an average of 10 random measurements.

The samples for mechanical analyses were prepared according to Polish standards (PN-EN ISO 527-1:2012) and determined using the TA-XT plus texture analyzer (Stable Micro Systems, Haslemere, UK). Composites were cut into 35 mm × 6 mm stripes and put into holders. The initial grip separation between holders was 20 mm and the rate of grip separation was 2 mm/min. Tensile strength (TS) was calculated by dividing tensile force (maximum force at rupture of the film) by the cross-sectional area of the film. The elongation percentage at break (E) was calculated by dividing the elongation at rupture by the initial gauge length and multiplying by 100. The reported results were the average values of ten replicates.

#### 2.3.6. Contact Angle Determination

Contact angles were determined using a Kruss-DSA100M (Kruss GmbH, Hamburg, Germany). Distilled water and pure diiodomethane’s contact angles were determined on the studied polysaccharide foil surfaces using the sessile drop method. The detailed methodology of the contact angle experiments and surface free energy analyses were presented in our previous paper [38]. We used the Owens-Wendt method [40], which is best for polymer property evaluation [41]. All measurements were performed in the environmental cell under constant temperature conditions (22 ± 0.3 °C) and humidity. For each foil sample, more than three successive tests were carried out.

#### 2.3.7. Bacteriostatic Activity Assay

Pure Culture Isolation and Identification of Tested Microorganisms

Samples of surface water (1000 mL) were collected from the Białka River (Bialka Tatrzanska, Poland) in sterile polypropylene bottles. The membrane filtration method was applied to isolate *Escherichia coli* (blue-green colonies on TBX agar, incubation at 44 °C, 48 h), while the serial dilution method was used to isolate mesophilic bacteria (Trypticase Soy Agar, 37 °C, 48 h) and *Staphylococcus* spp. (Chapman agar, 37 °C, 48 h). After incubation, the selected bacterial colonies were subcultured, subjected to Gram staining and the systematic positions of 17 colonies of presumptive *E. coli, Staphylococcus* spp. and *Bacillus* spp. were verified by MALDI-TOF (matrix-assisted laser desorption/ionization-time of flight) mass spectrometry.

Bacteriostatic Activity of Nanoparticles

In order to test the bacteriostatic activity of the examined nanoparticles, bacterial isolates were transferred into sterile saline solutions to prepare 0.5 MacFarland suspension standards, which were then streaked onto Mueller–Hinton II agar (BTL, Lodz, Poland). The 5-mm diameter disks of sterile foils were cut with a surface-sterilized scalpel and put onto the surface of the bacterial cultures. Foils without nanoparticles were used as a control. The plates were incubated at 35 °C for 18–24 h. After this time, the results were read, i.e., the diameters of inhibition of microbial growth around the nanoparticle foil discs were evaluated and measured. The larger the diameter of the inhibition zone, the stronger the bacteriostatic effect.

#### 2.3.8. Cell Culture

Human melanoma WM266-4 (ATCC^®^ CRL-1676™; malignant melanoma) cell lines were purchased from the American Type Culture Collections (ATCC, Manassas, VA, USA). Cells were cultured in controlled conditions (temp. 37 °C, 95% humidity, 5% CO_2_) according to the ATCC protocol in RPMI 1640 medium (Sigma-Aldrich, St. Louis, MO, USA) with the addition of 10% FBS (Sigma-Aldrich, St. Louis, MO, USA).

#### 2.3.9. Cell Treatment

Cells were seeded on the 96-well plates (8 × 103 cells per well). Twenty-four hours after seeding, the growth medium was replaced with a medium containing nanoparticle composites at concentrations ranging from 15 to 35%. Cells were treated for 24 and 48 h. Untreated cells in the growth medium were considered as a negative control (UC). Each treatment included three biological and three technical replicates.

#### 2.3.10. Cytotoxicity and Viability Assay

Concentrations of nanoparticle composites were selected based on their EC10 characteristics and confirmed using a Cytotoxicity LDH Test (Roche, Warsaw, Poland), according to the manufacturer’s protocols. Cell viability was determined by a Crystal Violet Assay (Sigma-Aldrich, St. Louis, MO, USA).

#### 2.3.11. Statistical Analysis

All experiments were performed at least three separate times and measured in triplicate. A Shapiro-Wilk’s test was applied to assess the normality of distribution. An independent samples 𝑡-test was applied to compare unpaired means between two groups. 𝑝 < 0.05 was considered statistically significant. All analyses were performed using Statistica ver.13.1 (StatSoft, Tulsa, OK, USA).

## 3. Results and Discussion

In order to confirm the presence of silver nanoparticles and determine their size and the dispersion of graphene within the bionanocomposite, we performed scanning electron microscopy using a SEM equipped with transmission electron (TED) and retractable backscattered-electron (RBEI) detectors. Figure 2 presents microscopy images of Hyal/GO and Hyal/Ag/GO nanocomposites. As shown in the figure, the presence of regularly spaced black sheets indicates that GO has been uniformly dispersed throughout the nanocomposite film and that GO does not aggregate in the polysaccharide matrix. We also observe the bubble-like surface of the Hyal/Ag/GO sample, which may be caused by the formation of nanocapsules (or nano-bubbles), sized 50–100 nm, during the generation of nanosilver due to the interaction of Hyal with nanosilver and graphene flakes [42].

We used a secondary electron detection (COMPO system), to assess the distribution of nanosilver in the composites—we observed that silver nanoparticles were located inside the bubbles and were evenly distributed. Using a TED detector, we managed to define the size and shape of silver nanoparticles. The obtained silver nanostructures were characterized by different sizes, ranging 10–20 nm, and they were regular, spherical, and the same as the silver nanoparticles obtained in Hyal matrix described in our previous papers [37,38] which means that the presence of GO did not affect the size or shape of silver nanoparticles.

Size exclusion chromatography results are presented in Table 1. We can observe an increase of the molecular weight in the samples containing nanoparticles. The highest increase was observed after the generation of Ag nanoparticles in the composite containing graphene oxide. We could also notice an increase in the radii of gyration in nanomaterial solutions compared to Hyal/C. This observation can be explained by interactions between hyaluronic acids, graphene oxide and nanometals.

The FTIR-ATR spectra in the spectral range of 700–4000 cm^−1^ for GO and Hyal/C, Hyal/Ag, Hyal/Ag/GO films are presented in Figure 3a. GO displayed characteristic FTIR peaks corresponding to its oxygen functionalities, including the C=O stretching vibration peak at 1731 cm^−1^, the C-O (epoxy) stretching vibration peak at 1227 cm^−1^, the C-O (alkoxy) stretching vibration peak at 1065 cm^−1^, and the vibration and deformation peaks of O-H groups at 3412 and 1627 cm^−1^, respectively.

The FTIR spectrum of Hyal showed the following characteristic bands: the peak located at 3240 cm^−1^ is associated with the intra-and intermolecular stretching vibration of the –OH group and the stretching vibration of the hydrogen bond from the –NH– group. The band at 2855 cm^−1^ can be attributed to the stretching vibration of the –CH_2_– group; the band at about 1601 cm^−1^ corresponds to the amide carbonyl and the band at 1402 cm^−1^ can be attributed to the stretching of the COO– group; the peak from 1023 cm^−1^ is attributed to the linkage stretching of C–OH. There are a very limited number of changes in the spectrum of Hyal/C and nanocomposite films. The Hyal/Ag/GO spectrum changes (decrease of the intensity of absorbance and no band shifts compared to Hyal/GO) suggest intermolecular interactions between the hydroxyl, carboxyl, amide carbonyl groups with silver nanoparticles and the absence of new covalent bonds.

The UV-Vis absorption spectra of Hyal/C, Hyal/Ag, Hyal/GO and Hyal/Ag/GO nanocomposites are presented in Figure 3b. There is a very large difference between the spectra of Hyal/C and the composites. GO spectrum presents a characteristic sharp absorption peak at about 233 nm and a broad shoulder at 290–305 nm, which is also clearly visible in the spectrum of Hyal/GO. The UV-Vis absorption spectra of the samples containing Hyal/Ag and Hyal/Ag/GO show an absorption band at 380–500 nm. These results point to the synthesis of Ag nanoparticles. The foils containing nanocomposites present higher absorbance compared to Hyal/C foil.

The effects of Ag and GO incorporation based on thickness, tensile strength (TS) and the elongation percentage at break (E) of hyaluronate composites are presented in Table 2. As was confirmed by statistical analysis, there were no differences in thickness between the composites. The tensile strength of Hyal composite was at a high level and was many times higher than composites made from other natural biodegradable polymers such as: starch [43,44], chitosan [45], cellulose acetate (CA) and cellulose nanofibers (CNF) [46].

The addition of both, nanosilver and graphene particles weakened the tensile strength of the Hyal composites, however, in the case of Ag the difference was statistically insignificant. In the presence of GO, this parameter decreased by 23.1%. The AG/GO combination had the most significant impact on the TS. According to Chen et al. [47], this fact may demonstrate the weak hydrogen-bonding interactions between biopolymer chains. Authors have claimed that, with enhanced hydrogen bonding between polymers molecular chains, remarkably, increased mechanical properties could be observed.

Despite the negative impact of Ag and GO, the results of the TS value were higher than those reported by Chen et al. [47] for chitosan/graphene oxide bionanocomposites (55.7 ± 1.0 MPa) or Chen et al. [48] for chitosan/alginate bionanocomposites enhanced with graphene oxide (29.1 ± 4.9 MPa). Additionally, obtained composites were mechanically stronger than commonly used plastic foils with LDPE (12.4–25.7 MPa) reported by Czarnecka-Komorowska et al. [49].

Ag, GO or Ag/GO introduction into the hyaluronic composite reduced its extensibility by 2 to 6 times. In this way, composites with greater stiffness were obtained. Nevertheless, the value of elongation at break for Hyal composites was higher than those reported by Chen et al. [47] regarding chitosan and chitosan/carboxymethyl cellulose films (22.6 and 10%). The same authors noted that bionanocomposites of chitosan polyelectrolyte complexed materials with graphene oxide (GO) had lower elongation (15.3%) than the control sample (chitosan matrix alone—22.6%).

The water contact angle analysis shows that the hyaluronic acid foils without any additive is the most hydrophobic sample (Table 3). The addition of graphene nanoparticles almost does not change the Hyal wettability. On the contrary, the presence of silver nanoparticles switches the sample hydrophobicity range to about 40–50°, which proves at least partial hydrophilic properties for such material. The data from diiodomethane contact angles do not show changes as pronounced the wetting angles tested with water. As seen, all measured values are in the range of 40–55°, and the overall trend of changes cannot be observed.

The further analysis of the surface free energy using the Ownes–Wendt method lets us conclude that graphene nanoparticle addition slightly diminishes hyaluronic acid foils’ total free surface energy. At the same time, the presence of silver nanoparticles significantly increases the total energy. However, the problem is more complex, and a detailed analysis of polar and dispersive energies must be performed and discussed. Hylauronic acid foils without any additives have the smallest polar energy. The addition of any of the particles increases polar energy in graphene slightly, while it does so significantly for silver. Simultaneously, hyaluronic foil without any particles has the highest dispersive energy, and the situation here is the opposite for polar energy. The addition of any of the particles diminishes the dispersive energy of hyaluronic acid foils. The most substantial effect is observed for foils dotted with only silver particles. In the presence of graphene (alone or in the co-composition with silver), the impact is not so large.

The bacteriostatic activity of different foils was tested with 17 environmental strains of bacteria from the following species: *S. saprophyticus* (*n* = 1), *S. equorum* (*n* = 1), *B. licheniformis* (*n* = 2), *B. subtilis* (*n* = 1) and *E. coli* (*n* = 12). The results of all individual readouts are shown in Table 4 and summarized in Figure 4. It was observed that nanoparticles Hyal/Ag and Hyal/Ag/GO caused growth inhibition of bacteria, but to various degrees, depending on the species and strain tested. No growth inhibition zone was observed in the case of Hyal/GO. In the case of Hyal/Ag, the highest mean growth inhibition was observed for *Staphylococcus* spp. (mean of 18 mm), followed by *Bacillus* spp. (mean of 16.3 mm) and *E. coli* (mean of 13.4 mm), while in the case of Hyal/Ag/GO the highest mean growth inhibition was observed for *Bacillus* spp. (mean of 15.3 mm), followed by *Staphylococcus* spp. (14.5 mm) and *E coli* (13.8 mm). These results suggest higher susceptibility of Gram-positive bacteria (i.e., *Staphylococcus* spp. and *Bacillus* spp.) to Hyal/Ag silver nanocomposites than the susceptibility of Gram-negative *E. coli*. The differences in the susceptibility proved to be statistically significant in the case of Hyal/Ag (*t*-test: *t* = 3.98; *p* = 0.0012).

The antimicrobial activity of silver has been attributed to certain morphological and physicochemical characteristics, including size, shape, colloidal stabilization, surface corona, composition, aggregation behavior, surface coating, surface/volume ratio, etc., which—when properly used—contribute to the inhibitory potential against several pathogenic microorganisms [50]. The antimicrobial (both antifungal and antibacterial) activity of silver nanoparticles synthesized using a number of various approaches has been demonstrated by, e.g., Abbasza-degan et al. [51], Guzman et al. [34], Khatoon et al. [35] and Khatoon et al. [36]. It was observed in some studies, that Gram-positive bacteria (e.g., *Staphylococcus* spp. or *Bacillus* spp.) are less susceptible to silver nanoparticles, compared to Gram-negative bacteria, such as *E. coli* [35,50]. The differences in the reaction of those two groups of bacteria to silver nanoparticles have been attributed to, e.g., greater thickness of peptidoglycan layer of Gram-positive bacteria or the presence of negative charge on the peptidoglycan layer that inactivates the bioactive Ag^+^ ions generated from silver nanoparticles [51]. However, growth inhibition zones in the studies by Guzman et al. [34], in which the effect of silver nanoparticles was tested against Gram-positive (*S. aureus*) and Gram-negative bacteria (*E. coli* and *P. aeruginosa*), were larger in the case of *S. aureus* bacteria than in the case of *E. coli* and *P. aeruginosa*, indicating higher susceptibility of the former ones to Ag NPs. These results are consistent with observations made in our study (Table 4 and Figure 5). According to Franci et al. [33], the most essential and most corroborated mechanism of action of Ag NPs against *E. coli* is the alteration of membrane permeability and respiration, while against *S. aureus*—irreversible damage to bacterial cells occurs. In view of the obtained results, it is reasonable to acknowledge that silver nanoparticles examined in this study are effective against *Bacillus* spp., *Staphylococcus* spp. And *E. coli*, which are potential human pathogens.

In order to investigate the effect of the obtained nanocomposites on human melanoma cells, we performed a Cytotoxicity LDH test and a Crystal Violet Assay. Crystal violet is a triarylmethane dye that binds to ribose type molecules such as DNA in nuclei. The number of the cells dyed with crystal violet is directly proportional to the number of attached cells to the plate, which assumes that they are “alive” The lactate dehydrogenase (LDH) test was performed to assess the toxicity of the tested substance to the cell culture [52]. LDH is a cytosol enzyme, which means that it is not being released to the environment in physiological conditions. When the cell membrane is destroyed or harm causes the death of a cell, it can be found outside the cell. Given that membrane permeability is a key feature of necrotic cells, measuring the release of LDH as a cytosolic enzyme, in combination with other methods, is a useful method for detecting necrosis.

In our study we used the LDH test and the Crystal Violet Assay, which gave us the influence of the nanoparticle composites on the decrease of cell viability excluding their potential necrosis. The results of the LDH test are presented in Table 5 and the cell viabilities are shown in Figure 5.

After 24 h, a 10–18% decrease in cell viability was observed for all samples containing nanoparticles compared to the negative control (NC) (100%) and to Hyal/C (Figure 5a). There was no difference between them on the viability of WM-266-4 melanoma cells. After 48 h, a significant decrease in cell viability was observed for all composite samples vs. NC and all results were statistically significant (Figure 5b). Analyzing the concentrations of nanocomposites used in the study, it can be concluded that with their increase the inhibition of WM-266-4 cell growth increased, with a particular reduction in viability (above 50%) at the concentrations of 30 and 35%. Among the nanoparticle composites used, the strongest inhibitory effect on the growth of WM-266-4 cells was demonstrated by Hyal/Ag/GO (Figure 5b), where the range of cell viability reduction ranged from 45% at the lowest concentration to 62% at the highest concentration of the used nanocomposite. In the case of Hyal/Ag/GO, the effect of all concentrations (15–35%) was also statistically significant vs. Hyal/C (Figure 5b). There was no release of LDH to the cellular medium which indicates no involvement of necrosis in the process of cell death under the influence of Hyal/Ag/GO (Table 5). In the case of Hyal/GO, the decrease in the number of cells was connected with an increase in the LDH concentration in the cell medium, which may indicate the involvement of necrosis in the process of cell death (Table 5). There was no cytonecrotic effect of the remaining Hyal/Ag and Hyal/C nanocomposites on WM-266-4 cells (Table 5).

## 4. Conclusions

The preparation of thin, elastic and transparent foils containing silver nanoparticles and graphene oxide in Hyal matrix was successfully carried out using an environmentally friendly method. In the composite containing GO and nanosilver, we could observe a formation of nanocapsules (or nano-bubbles) sized 50–100 nm with silver nanoparticles sized 10–20 nm regularly distributed inside the bubbles. The results of the structural analysis indicated intermolecular interactions of the functional groups in Hyal and GO with silver nanoparticles. The addition of GO did not improve the mechanical properties of Hyal foils. The results of the measurements of wettability and surface free energy confirmed that the addition of nanoparticles increased the polar energy and decreased the dispersion energy of the sample. Silver nanoparticles affected the sample much more than graphene particles, but when both graphene and silver particles were present in the system, we observed a synergistic effect from both components. Bacteriostatic activity was observed only in the case of Ag-containing nanocomposites, while those containing sodium hyaluronate or graphene/sodium hyaluronate were ineffective against the tested bacteria, i.e., *Bacillus* spp., *Staphylococcus* spp. and *E. coli*. The differences between Hyal/Ag and Hyal/Ag/GO were insignificant, contrary to their effectiveness against tested bacterial species and genera. Gram-positive bacteria (*Bacillus* spp. and *Staphylococcus* spp.) proved to be more susceptible to the silver nanocomposites than Gram-negative *E. coli*. The composites containing graphene oxide and nanosilver/graphene oxide exhibited cytotoxic effects on the tested WM-266-4 human melanoma cell line. The strongest inhibitory effect on cell growth was demonstrated by Hyal/Ag/GO and there was no involvement of necrosis in the process of cell death (which could be observed in the case of Hyal/GO). There was no cytonecrotic effect of the remaining Hyal/Ag and Hyal/C nanocomposites on the tested cells.

## Figures and Tables

**Figure 1 materials-14-03377-f001:**
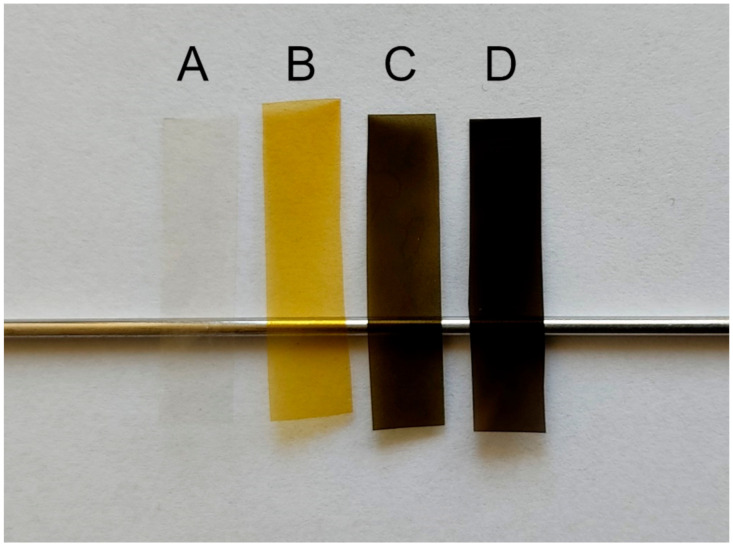
Obtained foils: A—Hyal/C; B—Hyal/Ag; C—Hyal/Ag/GO; D—Hyal/GO.

**Figure 2 materials-14-03377-f002:**
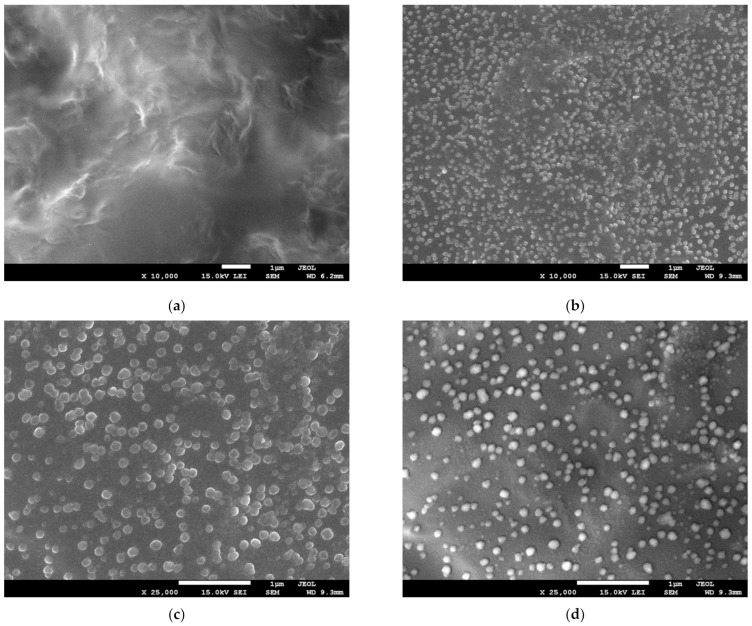
SEM (**a**–**d**) and TEM (**e**) micrographs of foils taken at different magnifications: (**a**) Hyal/GO; (**b**) Hyal/Ag/GO; (**c**) Hyal/Ag/GO at ×25,000 magnification; (**d**) Hyal/Ag/GO at ×25,000 magnification using secondary electron detection (COMPO system); (**e**) Hyal/Ag/GO at ×50,000 magnification.

**Figure 3 materials-14-03377-f003:**
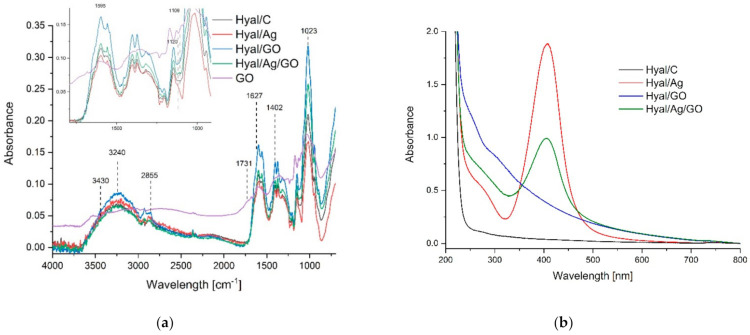
(**a**) FTIR spectra of Hyal/C (black line), Hyal/Ag (red line), Hyal/GO (blue line) and Hyal/Ag/GO (green line); (**b**) UV-vis spectra of Hyal/C (black line), Hyal/Ag (red line), Hyal/GO (blue line), Hyal/Ag/GO (green line) and GO (pink line).

**Figure 4 materials-14-03377-f004:**
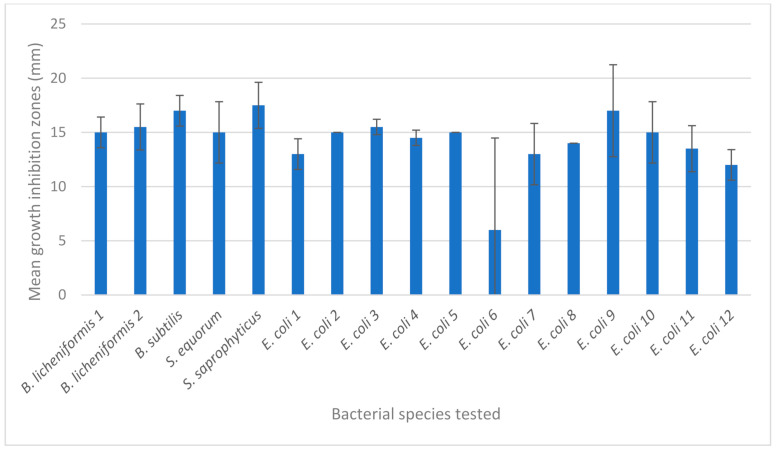
Mean growth inhibition zones (mm) caused by Hyal/Ag and Hyal/Ag/GO silver nanoparticle composites.

**Figure 5 materials-14-03377-f005:**
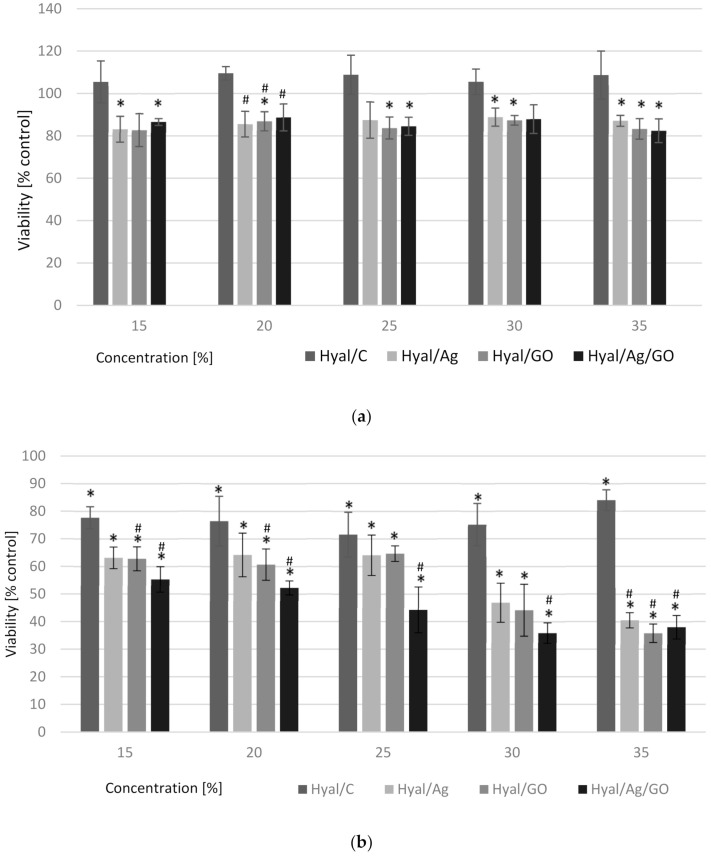
Effect of nanoparticle composites on WM-266-4 cells viability. WM-266-4 melanoma was seeded on the 96-well plates (8 × 103 cells per well). Twenty-four hours after seeding, the growth medium was replaced with a medium containing nanoparticle composites for (**a**) 24 h and (**b**) 48 h. Values are expressed as mean ± SD for *n* = 12, standardized to untreated control (UC) as 100%. Statistical significance was based on t-test * *p* ≤ 0.05 vs. UC and # *p* ≤ 0.05 vs. sodium hyaluronate.

**Table 1 materials-14-03377-t001:** The average molecular weight (Mw) and radius of gyration (Rg) of Hyal/C, Hyal/Ag, Hyal/GO, and Hy-al/Ag/GO.

Sample	Mw (Da)	Rg (nm)
Hyal/C	1.865 × 10^5^	49.4
Hyal/Ag	4.090 × 10^5^	58.5
Hyal/GO	4.768 × 10^5^	50.5
Hyal/Ag/GO	5.046 × 10^5^	52.8

**Table 2 materials-14-03377-t002:** Mechanical properties of composites.

Sample	Thickness (mm)	TS (MPa)	E (%)
Hyal/C	0.048 ± 0.011 ^a^	90.09 ± 9.52 ^a^	32.51 ± 6.94 ^a^
Hyal/Ag	0.051 ± 0.008 ^a^	81.27 ± 5.05 ^a^	14.14 ± 3.75 ^b^
Hyal/GO	0.049 ± 0.001 ^a^	69.30 ± 3.57 ^b^	13.99 ± 3.88 ^b^
Hyal/Ag/GO	0.050 ± 0.009 ^a^	57.80 ± 7.00 ^c^	5.11 ± 1.54 ^c^

Parameters in columns denoted with the same letters (a, b, c) do not differ statistically at the level of confidence *p* < 0.05.

**Table 3 materials-14-03377-t003:** The studied foils’ contact angle measured using water and diiodomethane (DIM) and the dispersive, polar and total surface free energies (calculated using the Owen–Wendt method).

	Water	DIM	Dispersive Energy (mJ/m^2^)	Polar Energy (mJ/m^2^)	Total Surface Free Energy (mJ/m^2^)
Hyal/C	84.0°	41.0°	41.59	2	43.60
Hyal/Ag	41.0°	52.0°	23.93	33.02	56.95
Hyal/GO	79.5°	55.3°	30.21	6.36	36.58
Hyal/Ag/GO	49.0°	43.5°	30.64	23.29	53.93

**Table 4 materials-14-03377-t004:** Growth inhibition of bacterial strains tests caused by the silver nanoparticle composites (mm). The results are means of three replicates.

Bacterial Strain	Nanoparticles			
	Hyal/Ag *	Hyal/GO	Hyal/Ag/GO	Hyal/C
*Bacillus licheniformis*	14	0	16	0
*Bacillus licheniformis*	17	0	14	0
*Bacillus subtilis*	18	0	16	0
*Staphylococcus equorum*	17	0	13	0
*Staphylococcus saprophyticus*	19	0	16	0
*Escherichia coli* 1	14	0	12	0
*Escherichia coli* 2	15	0	15	0
*Escherichia coli* 3	16	0	15	0
*Escherichia coli* 4	14	0	15	0
*Escherichia coli* 5	15	0	15	0
*Escherichia coli* 6	12	0	0	0
*Escherichia coli* 7	11	0	15	0
*Escherichia coli* 8	14	0	14	0
*Escherichia coli* 9	14	0	20	0
*Escherichia coli* 10	13	0	17	0
*Escherichia coli* 11	12	0	15	0
*Escherichia coli* 12	11	0	13	0
mean	14.47	0	14.18	0
standard deviation	2.35	0	4.07	0
coefficient of variation (%)	16.23	-	28.68	-

* The differences in growth inhibition zones between Gram-positive (*Bacillus* spp. and *Staphylococcus* spp.) and Gram-negative bacteria (*Escherichia coli*) obtained for Hyal/Ag are statistically significant (*t* = 3.39; *p* = 0.0012).

**Table 5 materials-14-03377-t005:** Cytotoxycity of nanoparticle composites on WM-266-4 cells.

		WM-266-4 Cytotoxicity %		
Concentration	Hyal/C ± SD	Hyal/Ag ± SD	Hyal/GO ± SD	Hyal/Ag/GO ± SD
24 h				
15%	0.35 ± 0.023	0.60 ± 0.043	2.78 ± 0.023	−1.34 ± 0.014
20%	0.84 ± 0.023	−3.01 ± 0.049	3.96 ± 0.023	−1.94 ± 0.048
25%	−3.60 ± 0.033	−2.70 ± 0.045	22.54 ± 0.023	−2.91 ± 0.044
30%	−3.59 ± 0.018	2.70 ± 0.049	12.73 ± 0.023	−1.12 ± 0.038
35%	5.44 ± 0.033	4.60 ± 0.025	22.50 ± 0.023	−3.26 ± 0.065
48 h				
15%	−9.39 ± 0.023	−3.56 ± 0.055	6.78 ± 0.033	−3.53 ± 0.025
20%	−9.81 ± 0.022	−3.71 ± 0.029	19.45 ± 0.060	−3.03 ± 0.017
25%	−10.75 ± 0.019	−3.99 ± 0.022	27.40 ± 0.027	−1.49 ± 0.069
30%	−12.03 ± 0.029	−4.15 ± 0.065	25.52 ± 0.015	−1.75 ± 0.040
35%	−11.07 ± 0.041	−4.58 ± 0.070	33.81± 0.062	−2.04 ± 0.074

WM-266-4 melanoma were seeded on the 96-well plates (8 × 10^3^ cells per well). Twenty-four hours later, growth medium was replaced with a medium containing nanoparticle composites. Cytotoxicity was measured with Cytotoxicity Detection Kit LDH (Roche, Poland). Values are expressed as mean ± SD for *n* = 12, standardized to untreated control (UC) as 100%.

## Data Availability

The data presented in this study are available on request from the corresponding author.

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
