# Peer review of "Formation and Investigation of Physicochemical, Biological and Bacteriostatic Properties of Nanocomposite Foils Containing Silver Nanoparticles and Graphene Oxide in Hyaluronic Acid Matrix"

_materials, 2021, doi:10.3390/ma14123377_

Round 1
Reviewer 1 Report
The basis of several discussion seems to be unconvincing for me. Further explanation will be helpful for readers.
Figure 3a: IR spectra have abundant information about conformation of materials. For examples, peak shifts with only several wavenumbers clearly reflect conformational changes of the molecules. The authors should indicate the values of wavenumber of the characteristic peaks for each spectrum in the figure to show basis of the argument written in line 314-315.
Line 315-317: The basis of this argument is unclear to me. Please write the reason why the authors concluded that GO interact with Hyal through intermolecular hydrogen bonds. Discussion about miscibility also seems to be unconvincing.
Line 395-397: As far as I can judge from table 4, it seems to be difficult to argue the higher growth inhibition for Gram-positive bacteria. Can the author provide statistical evidence for this statement?
Figure 4 seems to be not informative, because we can easily find these values in table 4. I recommend the authors to remove this figure.
Minor points
Line 81: et al.?
Line 303, 325: Figure 3a, 3b?
Figure 4, 5: Add y-axis label.
Figure 6: Add the values of concentration and viability for x- and y-axis.
Author Response
Dear Reviewer,
Thank you very much for your constructive comments. Based on your suggestions, we have made the corresponding revisions to our manuscript using track changes function in MS Word. The following is our itemized list of the changes that have been made.
Figure 3a: IR spectra have abundant information about conformation of materials. For examples, peak shifts with only several wavenumbers clearly reflect conformational changes of the molecules. The authors should indicate the values of wavenumber of the characteristic peaks for each spectrum in the figure to show basis of the argument written in line 314-315.
Thank you for this suggestion. We revised Fig. 3a – the several wavenumbers have been added. We also included IR spectrum for GO.
Line 315-317: The basis of this argument is unclear to me. Please write the reason why the authors concluded that GO interact with Hyal through intermolecular hydrogen bonds. Discussion about miscibility also seems to be unconvincing.
We changed the sentence to make it clearer
The Hyal/Ag/GO spectrum changes (decrease of the intensity of absorbance and no band shifts compared to Hyal/GO) suggest intermolecular interactions between the hydroxyl, carboxyl, amide carbonyl groups with silver nanoparticles and the absence of new covalent bonds.
Line 395-397: As far as I can judge from table 4, it seems to be difficult to argue the higher growth inhibition for Gram-positive bacteria. Can the author provide statistical evidence for this statement?
Values of the statistical t-test were provided, demonstrating the statistical significance of differences in the susceptibility of Gram-positive and Gram-negative bacteria for Hyal/Ag. The differences for Hyal/Ag/GO proved not to be statistically significant. The statement in lines 395-398 has been corrected accordingly.
Figure 4 seems to be not informative, because we can easily find these values in table 4. I recommend the authors to remove this figure.
Figure 4 was removed.
Minor points
Line 81: et al.? – we corrected the error
Line 303, 325: Figure 3a, 3b? – we corrected the error
Figure 4, 5: Add y-axis label. – we removed Figure 4 and added y-axis label to figure 5
Figure 6: Add the values of concentration and viability for x- and y-axis. – we added the missing values

Reviewer 2 Report
The manuscript titled “Formation and investigation of physicochemical, biological and bacteriostatic properties of nanocomposite foils containing silver nanoparticles and graphene oxide in the hyaluronic acid matrix” proposed a method to fabricate nano-foils using silver, graphene oxide, and hyaluronic acid. The authors did comprehensive studies in characterization and compatibility with cells. Overall, the manuscript is very well written. The authors may perform minor revisions to further improve the quality of this article.
- Table 1: I think the authors mean 105 (10^5) rather than 105?
- Line 325: I think the authors are referring to Figure 3b.
- For all figures and tables, if shortcuts are used, please provide full terms in the footnote of that figure/table.
- For the experiments of bacteriostatic activity of different foils, please include the data for the control group and mark “*” for statistically different groups similarly as Figure 6.
- Figure 6: please add the value for Y-axis.
Author Response
Dear Reviewer,
Thank you very much for your constructive comments. Based on your suggestions, we have made the corresponding revisions to our manuscript using track changes function in MS Word. The following is our itemized list of the changes that have been made.
The manuscript titled “Formation and investigation of physicochemical, biological and bacteriostatic properties of nanocomposite foils containing silver nanoparticles and graphene oxide in the hyaluronic acid matrix” proposed a method to fabricate nano-foils using silver, graphene oxide, and hyaluronic acid. The authors did comprehensive studies in characterization and compatibility with cells. Overall, the manuscript is very well written. The authors may perform minor revisions to further improve the quality of this article.
- Table 1: I think the authors mean 105 (10^5) rather than 105? - we corrected the error
- Line 325: I think the authors are referring to Figure 3b.- we corrected the error
- For all figures and tables, if shortcuts are used, please provide full terms in the footnote of that figure/table – we revised figures and tables according to the suggestion
- For the experiments of bacteriostatic activity of different foils, please include the data for the control group and mark “*” for statistically different groups similarly as Figure 6.
In Table 4 full names of bacterial genus and species were provided. In the case of Gram-positive/Gram/negative bacteria, t-test showed statistically significant differences in the case of Hyal/Ag and this was marked with an “*” in Table 4. In the case of summarized results, no statistically significant differences were demonstrated between individual species, therefore no “*” was placed in the graph.
- Figure 6: please add the value for Y-axis. - we added the missing values

Reviewer 3 Report
Reviewer report on Manuscript Draft entitled ‘Formation and investigation of physicochemical, biological and 2 bacteriostatic properties of nanocomposite foils containing sil- 3 ver nanoparticles and graphene oxide in hyaluronic acid ma- 4 trix’
In this manuscript, authors used prepared four types of foils containing pure sodium hyaluronate and silver nanoparticles (AgNPs), namely: (Hyal/C), nanosilver in Hyal matrix (Hyal/Ag), graphene oxide in Hyal matrix (Hyal/GO) and nanosilver with graphene oxide in Hyal matrix (Hyal/Ag/GO). The cytotoxic and antibacterial activity of these composite structures were examined.
Presented manuscript and discussions are rather valuable and interesting from the point of view of materials chemistry. The research is in scope of the journal. Therefore, the manuscript eventually can be published after some improvements and corrections:
Abstract is too long, it should be shortened, but anyway it should well address all most important findings.
Discussion of results should be significantly improved and significantly advanced. Results should be compared and discussed with that obtained by other authors in studies where cytotoxic activity and antibacterial properties of AgNPs are evaluated.
In addition to many others cytotoxicity experiments AgNPs were tested for antimicrobial activity against oral microorganismssuch as Staphylococcus spp, Bacillus spp. and E. coli. Therefore, in Introduction and Discussion parts other references on the evaluation of their antimicrobial activity of AgNPs-based particles (Comparative study of Antifungal Activity of Silver and Gold Nanoparticles Synthesized by Facile Chemical Approach. Journal of Environmental Chemical Engineering 2018. 6, 5837-5844. /// Antibacterial and antifungal activity of silver nanospheres synthesized by tri-sodium citrate assisted chemical approach. Vacuum 2017, 146, 259-265.) should be additionally overviewed and discussed.
Figure 4. is not meaningful – only two numbers, which can be presented as numbers in the text.
Conclusions are not very informative, therefore, they can be improved, advanced and probably a bit shortened.
Author Response
Dear Reviewer,
Thank you very much for your constructive comments. Based on your suggestions, we have made the corresponding revisions to our manuscript using track changes function in MS Word. The following is our itemized list of the changes that have been made.
In this manuscript, authors used prepared four types of foils containing pure sodium hyaluronate and silver nanoparticles (AgNPs), namely: (Hyal/C), nanosilver in Hyal matrix (Hyal/Ag), graphene oxide in Hyal matrix (Hyal/GO) and nanosilver with graphene oxide in Hyal matrix (Hyal/Ag/GO). The cytotoxic and antibacterial activity of these composite structures were examined.
Presented manuscript and discussions are rather valuable and interesting from the point of view of materials chemistry. The research is in scope of the journal. Therefore, the manuscript eventually can be published after some improvements and corrections:
Abstract is too long, it should be shortened, but anyway it should well address all most important findings.
Discussion of results should be significantly improved and significantly advanced. Results should be compared and discussed with that obtained by other authors in studies where cytotoxic activity and antibacterial properties of AgNPs are evaluated.
In addition to many others cytotoxicity experiments AgNPs were tested for antimicrobial activity against oral microorganisms such as Staphylococcus spp, Bacillus spp. and E. coli. Therefore, in Introduction and Discussion parts other references on the evaluation of their antimicrobial activity of AgNPs-based particles (Comparative study of Antifungal Activity of Silver and Gold Nanoparticles Synthesized by Facile Chemical Approach. Journal of Environmental Chemical Engineering 2018. 6, 5837-5844. /// Antibacterial and antifungal activity of silver nanospheres synthesized by tri-sodium citrate assisted chemical approach. Vacuum 2017, 146, 259-265.) should be additionally overviewed and discussed.
Thank you for this valuable remarks. The abstract and discussion have been revised. A paragraph was added to the introduction, reviewing the potential use of silver nanoparticles as potential antimicrobial agents. Also, the discussion section referring to antibacterial activity of silver nanoparticles was extended by referring to the studies where antibacterial and antifungal activity of silver nanoparticles obtained by various methods was tested. The two papers mentioned by the Reviewer have also been cited and discussed.
Figure 4. is not meaningful – only two numbers, which can be presented as numbers in the text.
We removed figure 4.
Conclusions are not very informative, therefore, they can be improved, advanced and probably a bit shortened.
The conclusions have been revised.
List of references added
- Kim, S-H., Lee, H-S, Ryu, D-S., Choi, S-J., Lee, D-S. (2011). Antibacterial activity of silver-nanoparticles against Staphylococcus aureus and Escherichia coli. Korean Journal of Microbiology and Biotechnology 39(1), 77-85.
- Lenart-Boroń, A. (2017). Antimicrobial resistance and prevalence of extended-spectrum beta-lactamase genes in Escherichia coli from major rivers in Podhale, southern Poland. International Journal of Environmental Science and Technology 14, 241-250.
- Umme Thahira Khatoon, G.V.S. Nageswara Rao, Mantravadi Krishna Mohan, Almira Ramanaviciene, Arunas Ramanavicius (2018) Comparative study of antifungal activity of silver and gold nanoparticles synthesized by facile chemical approach, Journal of Environmental Chemical Engineering, Volume 6, Issue 5, Pages 5837-5844
- Umme Thahira Khatoon, G.V.S. Nageswara Rao, Krishna M. Mohan, Almira Ramanaviciene, Arunas Ramanavicius (2017) Antibacterial and antifungal activity of silver nanospheres synthesized by tri-sodium citrate assisted chemical approach, Vacuum, Volume 146, Pages 259-265

Round 2
Reviewer 1 Report
The authors have answered to my all comments. Especially, the revision of figure 3a and statistical t-test seems to contribute to improvement of scientific soundness, but I want to make one comment about IR spectrum.
Line 320-323: The authors suggested presence of the intermolecular interaction based on limited spectrum changes. However, intermolecular interaction may also slightly change IR spectrum (broadening and band shifts). Therefore, I suppose that the limited changes of the spectrum suggest the absence of new covalent bonds and absence of strong intermolecular interaction, such as strong hydrogen bonding, rather than presence of intermolecular interaction between materials. Intermolecular interaction with silver nanoparticles may exist, but figure 3a seems to be not appropriate for basis of this argument.
Author Response
Thank you again for your time and valuable remarks. We analyzed the FTIR spectra again and inluded a magnified fragment of the spectrum and in our opinion there is a presence of electrostatic interractions or even complexation. We revised Fig. 3a.
The magnified fragments of the FTIR spectra are attached below. The bands at 1120 and 1109 cm-1 (Fig. 1) can be attributed (Fig. 1) to the interaction of the silver nanoparticles with the matrix [Bahrami, A., Mokarram, R. R., Khiabani, M. S., Ghanbarzadeh, B., & Salehi, R. (2019). Physico-mechanical and antimicrobial properties of tragacanth/hydroxypropyl methylcellulose/beeswax edible films reinforced with silver nanoparticles. International journal of biological macromolecules, 129, 1103-1112.] causing a kind of cross-linking, which corresponds with our molecular weight study (HPSE-MALLS) results.
Figure 1
Other authors (as for example cited below) obtained similar results:
- Ionita, M., Pandele, M. A., & Iovu, H. (2013). Sodium alginate/graphene oxide composite films with enhanced thermal and mechanical properties. Carbohydrate polymers, 94(1), 339-344.
- Kanmani, P., & Rhim, J. W. (2014). Physical, mechanical and antimicrobial properties of gelatin based active nanocomposite films containing AgNPs and nanoclay. Food Hydrocolloids, 35, 644-652.
On the other hand, the division of the band at 1595 cm-1 (Fig. 2) in the case of samples containing silver nanoparticles may also suggest the interaction of amide groups with silver nanoparticles. However, because of the relatively low concentration of silver nanoparticles in the composites, it is difficult to expect significant differences in the obtained spectra.
Figure 2
